# Design, Synthesis and Biological Evaluation of Novel Coumarin-Based Hydroxamate Derivatives as Histone Deacetylase (Hdac) Inhibitors with Antitumor Activities

**DOI:** 10.3390/molecules24142569

**Published:** 2019-07-15

**Authors:** Feifei Yang, Na Zhao, Jiali Song, Kongkai Zhu, Cheng-shi Jiang, Peipei Shan, Hua Zhang

**Affiliations:** 1School of Biological Science and Technology, University of Jinan, Jinan 250022, China; 2Institute for Translation Medicine, Qingdao University, Qingdao 266071, China

**Keywords:** coumarin, hydroxamate, HDAC inhibitors, antitumor growth, structure–activity relationship

## Abstract

A series of novel coumarin-based hydroxamate derivatives were designed and synthesized as histone deacetylase inhibitors (HDACis). Selective compounds showed a potent HDAC inhibition with nM IC_50_ values, with the best compound (**10e**) being nearly 90 times more active than vorinostat (SAHA) against HDAC1. Compounds **10e** and **11d** also increased the levels of acetylated histone H3 and H4, which is consistent with their strong HDAC inhibition. In addition, **10e** and **11d** displayed a higher potency toward human A549 and Hela cancer cell lines compared with SAHA. Moreover, **10e** and **11d** significantly arrested A549 cells at the G2/M phase and enhanced apoptosis. Molecular docking studies revealed the possible mode of interaction of compounds **10e** and **12a** with HDAC1. Our findings suggest that these novel coumarin-based HDAC inhibitors provide a promising scaffold for the development of new potential cancer chemotherapies.

## 1. Introduction

Coumarins, a class of plant secondary metabolites [1], are bicyclic heterocycles consisting of benzene and 2–pyrone rings and exhibit various pharmacological properties, including antidepressant [2], antibacterial [3], scavenging of reactive oxygen species (ROS) [4], anti-inflammatory [5], anticholinesterase [6], antithrombotic [7] and anticancer activities [8,9]. One of the most severe concerns in cancer treatment is related to the serious side effects of current chemotherapies. However, coumarins and their analogues exhibit a very rare cardiotoxicity, dermal toxicity, nephrotoxicity, hepatotoxicity and other adverse effects; therefore, they are generally considered as a group of promising candidates in anticancer drug discovery [10,11]. Coumarins influence a number of pathways in cancer, such as angiogenesis inhibition, kinase inhibition, carbonic anhydrase inhibition, cell cycle arrest, aromatase inhibition, antimitotic activity, telomerase inhibition, sulfatase inhibition and heat shock protein (HSP90) inhibition [1,12]. Numerous studies have confirmed the possible use of coumarins in cancer therapies. For instance, coumarin (1), 7–hydroxycoumarin (2), 6–nitro–7–hydroxycoumarin (3) and esculetin (4) (Figure 1) have been reported as potent anti-proliferative agents both in vitro and in xenograft models [13,14]. Because of the potential applications of coumarins in cancer treatment, extensive efforts have been made on the design and synthesis of coumarin derivatives with an improved anticancer activity.

Epigenetic changes in cancer are common and have been associated with pathogenesis and molecular heterogeneity [15]. Histone deacetylases (HDACs) are overexpressed in a variety of cancers and associated with tumorigenesis and development, thus making them attractive therapeutic targets [16,17]. The inhibition of HDACs has been identified as a promising therapeutic approach [18]. HDACis have been shown to significantly suppress cell proliferation, angiogenesis, metastasis and to lead to apoptosis through multiple mechanisms. These mechanisms include changes in the gene expression and alterations of both histone and nonhistone proteins, which are a remarkable biological phenomenon for the suppression of cancers [19,20,21].

To date, five HDACis, vorinostat (5, SAHA), romidepsin (6, FK228), belinostat (7, PXD101), panobinostat (8, LBH589) and chidamide (9, CS055) (Figure 1), have been approved for the treatment of cutaneous T cell lymphoma (CTCL), multiple myeloma (MM) or peripheral T cell lymphoma (PTCL), and over 20 other inhibitors are in different phases of clinical trials, which certifies that the development of novel HDACis is still one of the most important approaches for cancer treatment [22]. Despite their rich structural diversity, HDACis generally have a common pharmacophore model comprising of three motifs: a CAP region, a zinc-binding group (ZBG) and a linker part connecting CAP and ZBG. The development of HDACis mainly focuses on the modifications of the three parts, especially at the CAP and the linker regions, which define the activity and selectivity of inhibitors [23,24,25]. Nevertheless, current HDACis exhibit a poor efficacy against solid tumors, and thus, the development of effective HDACis that possess a high potency against solid tumors is still greatly needed [26].

HDACis have been employed in a number of efforts to generate hybrid agents with the purpose of achieving synergistic effects [27]. Several coumarin-based HDAC inhibitors (9a, 9b and 9c) (Figure 1) were designed and demonstrated the validity of such a design strategy to develop novel HDACis [28,29,30]. The only hydroxamate HDACi 9a was just synthesized as a fluorescent probe for determining the binding affinities (Kd) and the dissociation off-rates (Koff) of the HDAC-inhibitor complexes, while the antitumor activities associated with this compound were not studied. Therefore, in the present study, we incorporated coumarin into hydroxamate HDACis, and a series of new coumarin-based hydroxamate HDACis were then synthesized. The rationale for the design of these compounds is illustrated in Figure 2. We attempted the use of a substituted coumarin moiety as a CAP group for enzyme surface interactions, different carbon chains as a linker region, and a hydroxamic acid group as the ZBG. The synthesized compounds were evaluated for their inhibitory activities against HDAC1. The cytotoxicity against three different cancer cell lines, including A549 (adenocarcinoma human alveolar basal epithelial cells), Hela (human cervical cancer cell) and HepG2 (human liver cancer cells), were also conducted. Docking studies were performed to explore the interaction between selective compounds and HDAC1.

## 2. Chemistry

The routes used for the synthesis of the target compounds containing the coumarin moiety are shown in Scheme 1, Scheme 2 and Scheme 3. As depicted in Scheme 1, the intermediates **14a,b** were synthesized from 2–hydroxyacetophenone derivatives **13a,b** by reacting with diethyl carbonate and sodium hydride (NaH) at 100 °C [31]. After the chlorination, 4–chloro coumarins **15a,b** were further condensed with different amino acid esters in ethanol (EtOH) to yield esters **16a**–**e** according to procedure c, which were then reacted with freshly prepared hydroxylamine (NH_2_OH) in methanol (MeOH) to give the target compounds **10a**–**e**.

Compounds **11a**–**e** were prepared according to the methods described in Scheme 2. The intermediate **14b** was constructed from **13b**, which was alkylated with brominated fatty acid esters (n = 4**–**8) to yield ester compounds **17a**–**e**. Finally, treatment of **17a**–**e** with NH_2_OH in methanol gave the target compounds **11a**–**e**.

Compounds **12a**,**b** were obtained according to the methods described in Scheme 3. The substituted 7–hydroxycoumarins **18a**,**b** were reacted with Br(CH_2_)_7_CO_2_Me in the presence of potassium carbonate to produce esters **19a**,**b**, which were converted to hydroxamic acids **12a**,**b** with the treatment of NH_2_OH in MeOH.

## 3. Results and Discussion

### 3.1. HDAC Inhibition Assay

The HDAC inhibitory activities of the target compounds were assayed using HDAC1. The results were expressed as IC_50_ and summarized in Table 1, indicating that the HDAC1 inhibition activity was linker-length-dependent. The potency of the compounds increased with an increasing linker length. Compound **10e**, containing a seven-methylene linker, showed the best activity (IC_50_ = 0.24 nM) among its analogues and was 90 times more potent than SAHA (IC_50_ = 21.10 nM), which was consistent with the classical feature of HDACis [32,33]. It was also found that compounds with a 7–methoxy substitution on the coumarin moiety were slightly more potent than those without the substitution (**10b**
*vs.*
**10c**), which manifested the suitability of this substituent group for the HDAC inhibitory effect.

Upon further modification, the nitrogen atom was replaced with an oxygen atom, and a series of compounds **11a**–**e** were synthesized with different linker lengths in order to further testify the relationship between the linker length and HDAC1 inhibitory activities. As shown in Table 2, the activities of the target compounds were improved with the elongation of the linker. For example, compound **11d**, with *n* = 7, showed the most potent inhibitory activity. However, the inhibitory activity declined when *n* = 8. Then, for further modification, the seven-methylene linker was retained.

On the other hand, the linker and ZBG region were transferred to the C-7 position of the coumarin moiety, and compounds **12a**,**b** were synthesized. The inhibition activities of these two compounds were approximately three-times better than SAHA (Table 3). In addition, when the 3–hydrogen atom was replaced by methyl, the inhibitory activity was retained (**12a**
*vs.*
**12b**). However, the two compounds displayed decreased inhibitory activities compared with compound **11d**. The results showed that the substituted positions of the linker and ZBG region impacted the efficacy of the compounds.

### 3.2. IC_50_ Values of HDAC Isoforms Inhibition of Potent Compounds 

The selected compounds with a good HDAC1 inhibitory activity were also tested for their enzyme inhibitory activity against HDAC1, HDAC2, HDAC4, HDAC5, HDAC6, HDAC8, and HDAC11, in order to evaluate the selectivity of this series of compounds against HDAC isoforms (Table 4). The results displayed that compounds **10e** and **11d** were pan-HDAC inhibitors, which were similar to SAHA. The two compounds showed a more potent inhibition against class I and IIb HDAC isoforms than against class IV and IIa ones. Particularly for HDAC1, HDAC2 and HDAC8, **10e** exhibited about 88, 27 and 12 times, and **11d** showed about 11, 60 and 107 times greater potency, respectively, than SAHA. These results demonstrated that coumarin is an effective surface recognition cap for HDACs. 

### 3.3. Anti-Proliferative Activities against Three Cancer Cell Lines In Vitro

To investigate the anticancer activities, compounds **10e** and **11d** were then screened for their anti-proliferative activity against three cancer cell lines, and the IC_50_ values were summarized in Table 5. It was indicated that A549 and Hela cells were more sensitive to the selected compounds compared to HepG2 cells. Notably, compounds **10e** and **11d** exhibited comparable or better anti-proliferative activities when compared with SAHA against A549 and Hela cells.

### 3.4. Effects of Compounds **10e** and **11d** on Acetylated Histone Levels in A549 Cells

Based on the aforementioned results, we further investigated whether compounds **10e** and **11d** induced the acetylation of histones in lung cancer cells at different concentrations. A549 cells were incubated with the vehicle alone, and with SAHA, **10e** and **11d** (0.2, 0.5 and 1.0 μM) for 48 h, respectively. The levels of acetylated histone H3 and H4 were analyzed by Western blotting assays with β–actin as the negative control. The results in Figure 3A showed that compounds **10e** and **11d** could increase the expression of acetylated histone H3 and H4 in a dose-dependent manner. Meanwhile, Quantitative analysis results in Figure 3B showed that the levels of acetyl–histone H3 and H4 in compounds **10e** and **11d** treated groups were similar or even higher than those in the SAHA treated group (1.0 μM), which was consistent with their HDAC inhibition activities.

### 3.5. Compounds **10e** and **11d** Enhanced Apoptosis in the A549 Cell Line

To investigate whether the potent anti-proliferative activities of compounds **10e** and **11d** were related to enhancing the apoptosis of cancer cells, we carried out an annexin V fluorescein isothiocyanate (FITC)/propidium iodide (PI) binding assay in A549 cancer cells. As shown in Figure 4A, compounds **10e** and **11d** caused a significant induction of apoptosis in a dose-dependent manner. **10e** and **11d** induced 51.9% and 81.7% apoptosis in A549 cancer cells at 1.25 μM, induced 84.4% and 87.2% apoptosis in A549 cancer cells at 2.5 μM and induced 90.3% and 91.4% apoptosis in A549 cancer cells at 5 μM, respectively. However, SAHA only induced 27.6%, 40.9% and 61.4% of apoptotic cells at 1.25, 2.5 and 5 μM, respectively. The quantitative analysis results indicated that **10e** and **11d** were able to induce more apoptosis in A549 cells than the positive control drug, SAHA (Figure 4B).

### 3.6. Compounds **10e** and **11d** Induced Cell Cycle Arrest on A549 Cell Line

Next, we investigated whether the potent anti-proliferative activity of **10e** and **11d** resulted from the induction of the cell cycle arrest. As shown in Figure 5A and 5B, **10e** and **11d** led to a significant induction of the G2/M phase cell cycle arrest in a concentration-dependent manner. This was evidenced by the increasing percentages of cells in the G2/M phase, accompanied by a proportionate reduction in other phases of the cell cycle.

### 3.7. Theoretical Prediction of ADME Properties and Preliminary Toxicity Evaluation

For the purpose of achieving a better assessment of the druggability of these coumarin-based hydroxamate HDAC inhibitors, several parameters, including the calculated LogP (cLogP), topological polar surface area (tPSA), the number of hydrogen-bond acceptors and donors (n–ON and n–OHNH), and the number of rotable bonds and molecular volumes were carried out for the prediction of the ADME properties of the four compounds through the Molinspiration program (http://www.molinspiration.com/cgi-bin/properties) [34]. The reports suggested that compounds which meet the two criteria of (1) 10 or fewer rotatable bonds; (2) a polar surface area under 140 Å^2^ (or 12 or fewer H-bond donors and acceptors) may have a good oral bioavailability [35]. As shown in Table 6, compounds **10e** and **11d** both have 10 rotatable bonds, seven hydrogen bond acceptors (n–ON) and ≤ 3 donors (n–OHNH). The polar surface area (tPSA) did not exceed 140 A^2^. The cLogP was in an acceptable range (−2 to 5). The above results indicated that the two compounds are in a reasonable region for further development as potential drug candidates.

### 3.8. Molecular Docking Studies 

In order to determine the interaction mode between our compounds and HDACs, molecular docking was performed using a validated molecular dock program (AutoDock 4.27) [36]. Compound **10e** was docked into the active site of HDAC1 (PDB entry: 4BKX) (Figure 6). The results showed that **10e** had a similar binding mode to SAHA in the active site of HDAC1: the cap group interacts with the residues of the entrance region, the linker goes through the hydrophobic channel, and the hydroxamic acid could chelate the catalytic zinc ion (Figure 6A). Compound **10e** could form five hydrogen bonds with ASP99, HIS28, HIS178, ASP176 and ASP264 residues in the active site of HDAC1 (Figure 6B). It should be noted that the 7–methoxy group on the coumarin ring of compound **10e** docked into the hydrophobic pocket of the protein, which could account for the potency of this compound (Figure 6A).

## 4. Materials and Methods

### 4.1. Chemistry: General Methods 

The reagents were purchased from Energy Chemical Inc. Adamas-beta Ltd, J&K Inc. or Aladdin-reagents Inc., and were used without further purification unless otherwise specified. All reactions were carried out with the use of standard techniques under an inert atmosphere. ^1^H- and ^13^C-NMR spectra were generated on a Bruker 600 MHz instrument and obtained as CD_3_OD or DMSO-*d*_6_ solutions, using residual solvent peaks as references (δ_H_ 3.31 and δ_C_ 49.0 ppm for CD_3_OD; and δ_H_ 2.50 and δ_C_ 39.50 for DMSO-*d*_6_). Chemical shifts were shown in ppm and coupling constants (*J*) in Hz. Standard abbreviations indicating spin multiplicities are given as follows: s (singlet), d (doublet), t (triplet), q (quartet), br (broad) or m (multiplet). The high-resolution mass spectra were gathered on an Agilent 6545 Q-TOF mass spectrometer operating in electrospray ionization (ESI) mode. HPLC (Agilent Technologies 1260 Series) was employed for the purity determination, using the following method: Eclipse XDB C18 column, 5 μm, 4.6 mm × 150 mm, column temperature 40 °C, 1.5 mL/min MeOH-H_2_O system, 40%−90% in 10 min, hold on 5 min, and then back to 40% in 5 min.

### 4.2. General Procedures for the Preparation of Target Compounds

*N–hydroxy–3–((2–oxo–2H–chromen–4–yl)amino)propanamide* (**10a**). To a solution of hydroxyl amine hydrochloride (2.40 g, 34.8 mmol) in 10 mL MeOH, KOH (1.95 g, 34.8 mmol) was added. The reaction mixture was stirred for 10 min at 40 °C, and was then cooled to 0 °C and filtered. 3*–*((2*–*oxo*–*2H*–*chromen*–*4*–*yl)amino)propanoate methyl ester (320 mg, 1.1 mmol) was added to the filtrate followed by KOH (195 mg, 3.5 mmol), after which the reaction was stirred at room temperature for 30 min. The solvent was removed under reduced pressure, diluted with a saturated NH_4_Cl aqueous solution, and extracted with EtOAc. The organic layer was washed with brine and dried over Na_2_SO_4_. The resulting solution was evaporated under a reduced pressure and then purified by column chromatography [eluting with EtOAc followed by 10:1 CH_2_Cl_2_/MeOH] to give compound **10a** (105 mg, 34.5% yield). ^1^H-NMR (600 MHz, DMSO-*d*_6_) δ 10.50 (brs, 1H), 8.81 (brs, 1H), 8.00 (d, *J* = 8.4 Hz, 1H), 7.76–7.75 (m, 1H), 7.59 (dd, *J* = 6.6, 6.0 Hz, 1H), 7.33*–*7.30 (m, 2H), 5.19 (s, 1H), 3.49*–*3.47 (m, 2H), 2.36 (t, *J* = 6.6 Hz, 2H). ^13^C-NMR (150 MHz, DMSO-*d*_6_) δ 166.90, 161.49, 153.06, 152.87, 131.92, 123.32, 122.38, 116.96, 114.40, 81.57, 40.04, 30.92. HRMS (ESI): calcd for [C_12_H_12_N_2_O_4_ + H] ^+^ 249.0870, found 249.0882. HPLC purity: 99.6%, t_R_ = 2.2 min.

Compounds **10b**–**e**, **11a**–**e** and **12a,b** were prepared according to the procedure described for the preparation of compound **10a.**

*N–hydroxy–6–((2–oxo–2H–chromen–4–yl)amino)hexanamide* (**10b**) (31.4% yield). ^1^H-NMR (600 MHz, DMSO-*d_6_*) δ 10.37 (brs, 1H), 8.68 (brs, 1H), 8.08 (d, J = 7.8 Hz, 1H), 7.66*–*7.64 (m, 1H), 7.60*–*7.57 (m, 1H), 7.33*–*7.30 (m, 2H), 5.14 (s, 1H), 3.22 (q, J = 6.0 Hz, 2H), 1.97 (t, J = 7.8 Hz, 2H), 1.66*–*1.61 (m, 2H), 1.58*–*1.53 (m, 2H), 1.37*–*1.31 (m, 2H). ^13^C-NMR (150 MHz, DMSO-d_6_) δ 169.02, 161.58, 153.10, 153.08, 131.86, 123.26, 122.48, 116.95, 114.47, 81.16, 42.22, 32.23, 27.21, 26.19, 24.90. HR-MS (ESI): calcd for [C_15_H_18_N_2_O_4_ + H] ^+^ 291.1339, found 291.1344. HPLC purity: 99.8%, t_R_ = 4.1 min.

*N–hydroxy–6–((7–methoxy–2–oxo–2H–chromen–4–yl)amino)hexanamide* (**10c**) (28.9% yield). ^1^H-NMR (600 MHz, DMSO-*d*_6_) δ 10.41 (brs, 1H), 8.68 (brs, 1H), 8.05 (d, *J* = 8.4 Hz, 1H), 7.68*–*7.66 (m, 1H), 6.89 (dd, *J* = 8.4, 2.4 Hz, 1H), 6.84 (d, *J* = 2.4 Hz, 1H), 4.98 (s, 1H), 3.83 (s, 3H), 3.20*–*3.15 (m, 2H), 1.96 (t, *J* = 7.8 Hz, 2H), 1.63*–*1.58 (m, 2H), 1.55*–*1.50 (m, 2H), 1.34*–*1.30 (m, 2H). ^13^C-NMR (150 MHz, DMSO-*d*_6_) δ 169.04, 162.09, 161.94, 154.86, 153.49, 123.93, 111.04, 107.66, 100.82, 79.13, 55.77, 42.11, 32.24, 27.33, 26.20, 24.93. HRMS (ESI): calcd for [C_16_H_20_N_2_O_5_ + H] ^+^ 321.1445, found 321.1450. HPLC purity: 99.5%, t_R_ = 5.0 min.

*N–hydroxy–7–((7–methoxy–2–oxo–2H–chromen–4–yl)amino)heptanamide* (**10d**) (30.1% yield). ^1^H-NMR (600 MHz, DMSO-*d*_6_) δ 10.35 (brs, 1H), 8.67 (brs, 1H), 7.98 (d, *J* = 8.4 Hz, 1H), 7.54*–*7.52 (m, 1H), 6.91 (d, *J* = 8.4 Hz, 1H), 6.86*–*6.85 (m, 1H), 5.00 (s, 1H), 3.84 (s, 3H), 3.20*–*3.16 (m, 2H), 1.95 (t, *J* = 7.8 Hz, 2H), 1.63*–*1.58 (m, 2H), 1.52*–*1.48 (m, 2H), 1.37*–*1.32 (m, 2H), 1.31*–*1.28 (m, 2H). ^13^C-NMR (150 MHz, DMSO-*d*_6_) δ 169.07, 162.08, 161.88, 154.85, 153.43, 123.68, 111.05, 107.59, 100.82, 79.19, 55.74, 39.92, 32.21, 28.34, 27.46, 26.27, 25.05. HRMS (ESI): calcd for [C_17_H_22_N_2_O_5_ + H] ^+^ 335.1601, found 335.1607. HPLC purity: 99.1%, t_R_ = 6.4 min.

*N–hydroxy–8–((7–methoxy–2–oxo–2H–chromen–4–yl)amino)octanamide* (**10e**) (25.4% yield). ^1^H-NMR (600 MHz, DMSO-*d*_6_) δ 10.35 (brs, 1H), 8.67 (brs, 1H), 7.97 (d, *J* = 8.4 Hz, 1H), 7.53*–*7.51 (m, 1H), 6.91 (dd, *J* = 8.4, 2.4 Hz, 1H), 6.88*–*6.85 (m, 1H), 5.00 (s, 1H), 3.84 (s, 3H), 3.20*–*3.18 (m, 2H), 1.94 (t, *J* = 7.2 Hz, 2H), 1.62*–*1.59 (m, 2H), 1.50*–*1.48 (m, 2H), 1.34*–*1.30 (m, 4H), 1.26*–*1.23 (m, 2H). ^13^C-NMR (150 MHz, DMSO-*d*_6_) δ 168.97, 162.08, 161.87, 154.85, 153.42, 123.66, 111.06, 107.59, 100.83, 79.19, 55.74, 39.92, 32.18, 28.55, 28.48, 27.56, 26.47, 25.06. HRMS (ESI): calcd for [C_18_H_24_N_2_O_5_ + H]^+^ 349.1758, found 349.1747. HPLC purity: 99.4%, t_R_ = 8.0 min.

*N–hydroxy–5–((7–methoxy–2–oxo–2H–chromen–4–yl)oxy)pentanamide* (**11a**) (31.2% yield). ^1^H-NMR (600 MHz, CD_3_OD) δ 7.78 (d, *J* = 8.4 Hz, 1H), 6.93 (dd, *J* = 8.4, 2.4 Hz, 1H), 6.88 (d, *J* = 2.4 Hz, 1H), 5.65 (s, 1H), 4.21 (t, *J* = 6.6 Hz, 2H), 3.88 (s, 3H), 2.20 (t, *J* = 7.2 Hz, 2H), 1.94*–*1.90 (m, 2H), 1.88*–*1.84 (m, 2H). ^13^C-NMR (150 MHz, CD_3_OD) δ 172.48, 168.23, 165.97, 165.16, 156.42, 125.35, 113.63, 110.05, 101.40, 88.40, 70.28, 56.41, 33.21, 29.04, 23.23. HRMS (ESI): calcd for [C_15_H_17_NO_6_ + H] ^+^ 308.1129, found 308.1122. HPLC purity: 98.9%, t_R_ = 5.8 min.

*N–hydroxy–6–((7–methoxy–2–oxo–2H–chromen–4–yl)oxy)hexanamide* (**11b**) (41.6% yield). ^1^H-NMR (600 MHz, DMSO-*d*_6_) δ 10.36 (brs, 1H), 8.69 (brs, 1H), 7.69 (d, *J* = 8.4 Hz, 1H), 6.98 (d, *J* = 2.4 Hz, 1H), 6.95 (dd, *J* = 8.4, 2.4 Hz, 1H), 5.73 (s, 1H), 4.18 (t, *J* = 6.6 Hz, 2H), 3.86 (s, 3H), 1.99 (t, *J* = 7.2 Hz, 2H), 1.83*–*1.78 (m, 2H), 1.61*–*1.56 (m, 2H), 1.46*–*1.42 (m, 2H). ^13^C-NMR (150 MHz, DMSO-*d*_6_) δ 168.96, 165.40, 162.88, 162.10, 154.63, 123.93, 112.19, 108.35, 100.53, 87.88, 69.20, 55.92, 32.14, 27.17, 25.03, 24.77. HRMS (ESI): calcd for [C_16_H_19_NO_6_ + H] ^+^ 322.1285, found 322.1285. HPLC purity: 99.2%, t_R_ = 7.1 min.

*N–hydroxy–7–((7–methoxy–2–oxo–2H–chromen–4–yl)oxy)heptanamide* (**11c**) (39.7% yield). ^1^H-NMR (600 MHz, CD_3_OD) δ 7.77 (d, *J* = 8.4 Hz, 1H), 6.95 (dd, *J* = 8.4, 2.4 Hz, 1H), 6.90 (d, *J* = 2.4 Hz, 1H), 5.66 (s, 1H), 4.21 (t, *J* = 6.6 Hz, 2H), 3.89 (s, 3H), 2.12 (t, *J* = 7.2 Hz, 2H), 1.94*–*1.90 (m, 2H), 1.70*–*1.65 (m, 2H), 1.59*–*1.54 (m, 2H), 1.47*–*1.42 (m, 2H). ^13^C-NMR (150 MHz, CD_3_OD) δ 168.34, 166.04, 165.17, 156.44, 125.31, 113.65, 110.11, 101.44, 88.33, 70.76, 56.42, 33.64, 29.76, 29.47, 26.72, 26.62. HRMS (ESI): calcd for [C_17_H_21_NO_6_ + H] ^+^ 336.1442, found 336.1433. HPLC purity: 96.9%, t_R_ = 18.8 min. 

*N–hydroxy–8–((7–methoxy–2–oxo–2H–chromen–4–yl)oxy)octanamide* (**11d**) (45.8% yield). ^1^H-NMR (600 MHz, CD_3_OD) δ 7.75 (d, *J* = 8.4 Hz, 1H), 6.94 (dd, *J* = 8.4, 2.4 Hz, 1H), 6.88 (d, *J* = 2.4 Hz, 1H), 5.65 (s, 1H), 4.20 (t, *J* = 6.6 Hz, 2H), 3.89 (s, 3H), 2.10 (t, *J* = 7.2 Hz, 2H), 1.92–1.89 (m, 2H), 1.67*–*1.62 (m, 2H), 1.58*–*1.52 (m, 2H), 1.46*–*1.43 (m, 2H), 1.40*–*1.37 (m, 2H). ^13^C-NMR (150 MHz, CD_3_OD) δ 172.98, 168.33, 166.04, 165.15, 156.41, 125.27, 113.63, 110.10, 101.44, 88.32, 70.84, 56.42, 33.72, 29.99, 29.98, 29.54, 26.89, 26.65. HRMS (ESI): calcd for [C_18_H_23_NO_6_ + H] ^+^ 350.1598, found 350.1596. HPLC purity: 99.2%, t_R_ = 10.3 min. 

*N–hydroxy–9–((7–methoxy–2–oxo–2H–chromen–4–yl)oxy)nonanamide* (**11e**) (42.3% yield). ^1^H-NMR (600 MHz, DMSO-*d*_6_) δ 10.33 (brs, 1H), 8.66 (brs, 1H), 7.69 (d, *J* = 8.4 Hz, 1H), 6.98 (d, *J* = 2.4 Hz, 1H), 6.96 (dd, *J* = 8.4, 2.4 Hz, 1H), 5.73 (s, 1H), 4.18 (t, *J* = 6.6 Hz, 2H), 3.85 (s, 3H), 1.94 (t, *J* = 7.2 Hz, 2H), 1.81*–*1.79 (m, 2H), 1.50*–*1.47 (m, 2H), 1.46*–*1.43 (m, 2H), 1.35*–*1.32 (m, 2H), 1.30*–*1.26 (m, 2H), 1.26*–*1.22 (m, 2H). ^13^C-NMR (150 MHz, DMSO-*d*_6_) δ 169.07, 165.41, 162.87, 162.09, 154.62, 123.89, 112.20, 108.36, 100.54, 87.85, 69.29, 55.91, 32.23, 28.64, 28.56, 28.52, 27.95, 25.38, 25.08. HRMS (ESI): calcd for [C_19_H_25_NO_6_ + H] ^+^ 364.1755, found 364.1761. HPLC purity: 99.0%, t_R_ = 18.7 min. 

*N–hydroxy–8–((2–oxo–2H–chromen–7–yl)oxy)octanamide* (**12a**) (37.6% yield). ^1^H-NMR (600 MHz, DMSO-*d*_6_) δ 10.34 (brs, 1H), 8.67 (brs, 1H), 7.99 (d, *J* = 9.0 Hz, 1H), 7.62 (d, *J* = 8.4 Hz, 1H), 6.98 (d, *J* = 2.4 Hz, 1H), 6.94 (dd, *J* = 8.4, 2.4 Hz, 1H), 6.28 (d, *J* = 9.0 Hz, 1H), 4.06 (t, *J* = 6.6 Hz, 2H), 1.94 (t, *J* = 7.2 Hz, 2H), 1.74*–*1.71 (m, 2H), 1.51*–*1.48 (m, 2H), 1.42*–*1.39 (m, 2H), 1.33*–*1.29 (m, 2H), 1.28*–*1.24 (m, 2H). ^13^C-NMR (150 MHz, DMSO-*d*_6_) δ 169.08, 161.88, 160.32, 155.42, 144.35, 129.46, 112.71, 112.36, 112.22, 101.10, 68.25, 32.22, 28.49, 28.40, 28.38, 25.31, 25.04. HRMS (ESI): calcd for [C_17_H_21_NO_5_ + H] ^+^ 320.1492, found 320.1488. HPLC purity: 99.1%, t_R_ = 8.9 min. 

*N–hydroxy–8–((4–methyl–2–oxo–2H–chromen–7–yl)oxy)octanamide* (**12b**) (33.2% yield). ^1^H-NMR (600 MHz, DMSO-*d*_6_) δ 10.33 (brs, 1H), 8.66 (brs, 1H), 7.67 (d, J = 9.0 Hz, 1H), 6.96–6.94 (m, 2H), 6.20 (s, 1H), 4.07 (t, J = 6.6 Hz, 2H), 2.40 (s, 3H), 1.94 (t, J = 7.2 Hz, 2H), 1.75*–*1.70 (m, 2H), 1.52*–*1.47 (m, 2H), 1.43*–*1.38 (m, 2H), 1.35*–*1.30 (m, 2H), 1.28*–*1.23 (m, 2H). ^13^C-NMR (150 MHz, DMSO-*d*_6_) δ169.09, 161.79, 160.19, 154.77, 153.45, 126.44, 112.99, 112.43, 111.04, 101.12, 68.23, 32.22, 28.50, 28.41, 28.39, 25.32, 25.04, 18.13. HRMS (ESI): calcd for [C_18_H_23_NO_5_ + H] ^+^ 334.1649, found 334.1637. HPLC purity: 98.8%, t_R_ = 10.3 min. 

Compounds **10a**–**e** and **11a**–**e** have been published in patent no. CN 108658915A (Oct 16, 2018) [37].

### 4.3. HDAC1 Inhibitory Assay

The HDAC1 enzyme activity in vitro was determined by the protease-coupled assay. Different concentrations of compounds were incubated with recombinant HDAC1 (BPS Biosciences, San Diego, CA, USA) at room temperature for 15 min, followed by adding trypsin, as well as Ac-peptide-AMC substrates, to initiate the reaction in a Tris-based assay buffer. Fluorescent AMC, released from the substrate, was measured in SynergyMx (BioTek, VT, USA) using a filter set at excitation = 355 nm and emission = 460 nm. The IC_50_ values were calculated by GraphPad Prism software (La Jolla, CA, USA). (for details see Appendix A)

### 4.4. In Vitro Anti-Proliferation Assay. 

The in vitro anti-proliferation assay was operated as previously described [38]. In brief, the human cancer cell lines, A549, HeLa and HepG2, were seeded into 96-well plates at the appropriate cell densities. After incubation for 24 h, the cells were treated with various concentrations of tested compounds for 48 h. Then, the 3*–*(4,5*–*dimethyl*–*2-thiazolyl)*–*2,5*–*diphenyl*–*2*–*H*–*tetrazolium bromide (MTT) solution was added and co-incubated for another 4 h. The reactions were stopped by the addition of DMSO solution, and the samples were measured at 490 nm by a microplate spectrophotometer (Thermo Fisher Scientific, Waltham, MA USA). The IC_50_ was calculated using GraphPad Software. Three independent experiments were carried out in triplicate. (for details see Appendix A)

### 4.5. Western Blot Analysis

Western blotting was performed as previously described [39]. Briefly, human cancer cells were lysed in a lysis buffer (50 mM Tris-HCl pH 8.0, 5 mM EDTA, 100 mM NaCl, 0.5% NP-40, 1 mM PMSF), centrifuged for 10 min at 10,000 g, and the insoluble debris were discarded. Cell lysates were further analyzed with SDS-PAGE and western blotting with the indicated antibodies.

### 4.6. Cell Apoptosis Analysis

The cell apoptosis analysis was measured by annexin V FITC/PI assay using the Vybrant Apoptosis Assay kit (Invitrogen, Carlsbad, CA, USA). Briefly, A549 cells (8 × 10^4^/well) were treated with DMSO and compounds **10e** or **11d** for 72 h. The cells were then harvested and stained with annexin-binding buffer, Alexa Fluor 488 annexin and propidium iodide for 15 min in the dark. After staining, 400 μL of 1X annexin-binding buffer was added, mixed gently and kept on ice. The samples were measured using a BD Biosciences FACS Aria flow cytometer.

### 4.7. Cell Cycle Analysis

The cell cycle analysis was carried out by estimating the DNA contents with flow cytometry. After incubation with the indicated doses of **10e** or **11d** for 24 h, A549 cells were fixed in 70% ice-cold ethanol, incubated overnight at –20 °C, stained with propidium iodide (PI)/Triton X-100 containing RNaseA solution for 20 min at 37 °C, and then analyzed by FACS.

### 4.8. Molecular Docking Studies

Molecular docking studies were carried out with Autodock-4.27. For the docking calculations, The HDAC1 crystal structure (PDB code: 4BKX) was retrieved from the Protein Data Bank (www.pdb.org). Before docking, all the water molecules were removed from HDAC1, and Gasteiger partial charges were assigned to the selected compounds and enzyme atoms. The docking results were analyzed with the programs AutoDockTools, 27 DOCKRES and VMD.

## 5. Conclusions

Coumarins have been identified as anticancer candidates. HDACis are one of the hot topics in the field of antitumor research. In order to achieve an increased anticancer efficacy, a series of hybrid compounds bearing coumarin and hydroxamic acid scaffolds have been designed and synthesized as novel HDACis, and their biological activities have been evaluated in a series of *in vitro* assays. Compound **10e** showed the most potent inhibitory activity against HDAC1, with IC_50_ of 0.24 nM, which was almost 90 times lower than SAHA (IC_50_ = 21.10 nM). *In vitro* cell growth inhibition assays displayed that compound **10e** and **11d** exhibited better inhibitory activities against the human lung cancer cell line A549 and cervical cancer cell line Hela than against the hepatocellular carcinoma cell line HepG2. A western blotting analysis revealed that compounds **10e** and **11d** could upregulate the levels of acetylated histone H3 and H4 in a dose-dependent manner. In addition, compound **10e** and **11d** were found to induce A549 cancer cell cycle arrest and enhance cancer cell apoptosis. Furthermore, a molecular docking analysis exhibited a possible interaction mode of compounds **10e** and **12a** with HDAC1. Taken together, all these data suggest that these novel coumarin-based HDAC inhibitors could be promising candidates for the further development of novel antitumor agents.

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
