# Peer review of "Design, Synthesis and Biological Evaluation of Novel Coumarin-Based Hydroxamate Derivatives as Histone Deacetylase (Hdac) Inhibitors with Antitumor Activities"

_molecules, 2019, doi:10.3390/molecules24142569_

Round 1
Reviewer 1 Report
This is an interesting manuscript with novel data. The authors designed and synthesized novel HDAC inhibitors able that were evaluated in cancer cell lines.
The authors used several techniques including HDAC inhibitory assay, anti-proliferative activities, apoptosis analysis, western blot analysis of HDAC, cell cycle analysis and molecular docking studies. The authors concluded that the novel HDAC inhibitors may be useful for cancer chemotherapy.
Major points
- MTT is not useful to measure the anti-proliferative activities. According to reference [37] “cell proliferation of H1299 cells … was measured using a EdU incorporation assay” (legend to Fig 5 of reference 37). EdU= 5-Ethynyl-2-deoxyuridine. The author should conduct these experiments using an appropriate reagent.
- All the IC50 values should have a measure of dispersion (SD or SEM) since these values were obtained from several measurements. In addition, graphs from which these values were obtained must be included (for example, supplementary material)
- Tables with quantitative data (with a measure of dispersion) from apoptosis analysis, western blot analysis of HDAC and cell cycle analysis including statistical analysis should be included.
- A section of reagents containing the commercial providers should be included.
Minor points:
All the abbreviations should be defined the first time and then they should be used consistently (both in the abstract and in the main text). Examples: Line 3: HDAC; Line 15: SAHA; Line 176: FITC; Line 356: MTT.
A space should be inserted in some words: Example: line 176: “propidiumiodide”
A space should be inserted between the text and the whole tables. Example: between lines 118 and 119 and between lines 121 and 122. In the present manuscript is difficult to distinguish tables form text.
Line 351: “(California, US)” is incomplete. These data should contain the city and complete country “USA”
Reviewer 2 Report
In the manuscript, the authors have reported the synthesis of several histone deacetylase inhibitors, where some of the compounds shoed significant inhibition toward human and hela cancer cell lines in comparison with SAHA. The authors further show results of docking studies to understand the interaction of the compounds with HDAC1. The manuscript can be accepted after the authors the provide full forms for all abbreviation used and IUPAC names of all compounds mentioned so that the paper is not lost amongst the general audience (eg: SAHA, NH2OH, MeOH. etc).
Reviewer 3 Report
Manuscript ID molecules-524939 entitled "Design, synthesis and biological evaluation of novel coumarin-based hydroxamate derivatives as HDAC inhibitors with antitumor activities", by Yang F. et al is a well written work on the synthesis, biological and docking study of a set of coumarines derivatives acting as HDAC1 inhibitors. Surely results are very interesting, however authors have published them as a patent yet (no. CN 108658915, Oct 16, 2018). Only molecules 12a and 12b are really new, as well as the docking study, although this latter is not well described and is poorly informative. Therefore I suggest to reject this manuscript.
Round 2
Reviewer 1 Report
The authors performed the changes requestes by this reviewer
Author Response
We have checked our manuscript using the professional English editing service provided by MDPI. (the details were shown in the revised manuscript).
Reviewer 3 Report
For sure, I agree with Authors about patent purposes. However, in my opinion, this submission is a copy of the patented work and, therefore, it does not show nothing new.
Author Response
We appreciate the reviewer’s comment. On the one hand, we do have reported most of the compounds in the patent, however only partial bio-activity results were provided in the patent. In the manuscript, we reported the design idea of such series of compounds, other important experimental results (cell cycle, cell apoptosis) as well as the results of molecular docking studies for active compounds. So this submission was not a simple replication of the patented work. On the other hand, we applied for patent and submitted this article around the same time, it just took a little too long to edit the article, so in the future work, we will pay due attention to this problem.